# Crosstalk between the Rod Outer Segments and Retinal Pigmented Epithelium in the Generation of Oxidative Stress in an In Vitro Model

**DOI:** 10.3390/cells12172173

**Published:** 2023-08-30

**Authors:** Silvia Ravera, Nadia Bertola, Alessandra Puddu, Silvia Bruno, Davide Maggi, Isabella Panfoli

**Affiliations:** 1Department of Experimental Medicine, Università di Genoa, Via De Toni 14, 16132 Genova, Italy; 2Molecular Pathology Unit, IRCCS Ospedale Policlinico San Martino, Largo Rosanna Benzi 10, 16132 Genova, Italy; 3Department of Internal Medicine and Medical Specialties, University of Genova, Viale Benedetto XV 6, 16132 Genova, Italy; 4Department of Pharmacy-(DIFAR), Università di Genova, Viale Benedetto XV 3, 16132 Genova, Italy

**Keywords:** retinal pigmented epithelium, rod outer segments, aerobic metabolism, oxidative stress, lipofuscin, antioxidants, diabetic retinopathy, hyperglycemia

## Abstract

Dysfunction of the retinal pigment epithelium (RPE) is associated with several diseases characterized by retinal degeneration, such as diabetic retinopathy (DR). However, it has recently been proposed that outer retinal neurons also participate in the damage triggering. Therefore, we have evaluated the possible crosstalk between RPE and photoreceptors in priming and maintaining oxidative damage of the RPE. For this purpose, we used ARPE-19 cells as a model of human RPE, grown in normal (NG, 5.6 mM) or high glucose (HG, 25 mM) and unoxidized (UOx) or oxidized (Ox) mammalian retinal rod outer segments (OSs). ARPE-19 cells were efficient at phagocytizing rod OSs in both NG and HG settings. However, in HG, ARPE-19 cells treated with Ox-rod OSs accumulated MDA and lipofuscins and displayed altered LC3, GRP78, and caspase 8 expression compared to untreated and UOx-rod-OS-treated cells. Data suggest that early oxidative damage may originate from the photoreceptors and subsequently extend to the RPE, providing a new perspective to the idea that retinal degeneration depends solely on a redox alteration of the RPE.

## 1. Introduction

The retinal pigment epithelium (RPE) of polarized, melanosome-rich epithelial cells is located between the retinal photoreceptor layer and Bruch’s membrane, next to the choroid [1]. RPE impairment is linked to the development of degenerative retinal diseases such as diabetic retinopathy (DR) [2,3] and age-related macular degeneration (AMD) [4], both leading causes of blindness [2]. The interaction of the photoreceptors with the RPE, the Bruch’s membrane, and the choroid regulates the renewal of the 11-cis retinal and the outer segment (OS) of photoreceptors, the exchange of substances between the neural retina and the circulation and forms the outer blood–retinal barrier (OBRB) [5,6,7]. Rods represent most parts of the mammalian retina and the human parafovea [8]. The rod OS is a modified cilium involved in visual signal transduction [9]. In particular, the OS contains about 2000 stacked membranous disks that undergo a process of continuous renewal regulated by light [10]. While RPE continually phagocytes the OS tip disks, new disks are formed at the cilium base [11]. OS phagocytosis involves αvβ5 integrin receptors, focal adhesion kinase (FAK), and Mer tyrosine kinase (MerTK) [12,13,14]. The disk formation, still incompletely understood, involves the synthesis of a basal membrane evaginating next to the cilium to form sacs and the incorporation of disk proteins synthesized in the inner segment via intraflagellar transport proteins (IFT) [15,16]. Disk shedding has been proposed to also serve as an antioxidant system to ease the oxidative burden of the retina, typically restricted to the outer part [11]. The retina is vulnerable to oxidative stress due to its high oxygen (O_2_) consumption [17] and metabolic rates [18] and content of polyunsaturated fatty acids (PUFAs) [19]. Oxidative stress, a condition arising from an imbalance between reactive oxygen intermediate (ROI) production and scavenging [20], plays a pivotal role in the pathogenesis of retinal degenerative diseases, causing RPE and photoreceptor cell loss [3,21,22]. Apoptosis of rods, which display a preferential vulnerability over cones [8], causes, in turn, cone loss, impairing high-acuity vision. In fact, rods secrete a trophic cone viability factor, identified in a truncated thioredoxin-like protein [23]. Contrary to prior belief, the onset of DR has been lately identified in the oxidative damage of the neural retina that precedes retinal microvascular histological damage [24,25]. In experimental DR, the outer retina is the primary site of oxidative stress [25]. On the other hand, a spectroscopic method has localized such retinal oxidative stress at the interphase between the OS and the RPE [25], and mitochondrial dysfunction seems involved in DR onset [26]. To elucidate the pathogenesis of DR, it appears fundamental to understand the interactions between the RPE and the rod OS, considering that to sustain the energy demand of the phototransduction, the OSs display their own oxidative metabolism [27,28], which represents a source of oxidative stress inside the OS [29,30]. As no studies have investigated this topic, the present work aims to assess whether crosstalk exists between the OS and the RPE, causing cumulative oxidative damage to the outer retina. Thus, herein, we have investigated the effect of unoxidized (UOx) or oxidized (Ox) rod OS phagocytosis on ARPE-19 cells grown in normal- (NG) or high-glucose (HG) conditions. The ARPE-19 cells are a spontaneously immortalized cell line of human origin, which express RPE markers and display physiologically relevant features, such as barrier formation and the ability to phagocytize the rod OS [31]. ARPE-19 cells are considered an ex vivo model of human RPE [32] used to study several pathological changes associated with DR [33]. Intracellular trafficking and oxidation markers were studied to test whether phagocytosis of an Ox-rod OS could recapitulate the characteristics of the RPE pathology associated with DR.

## 2. Materials and Methods

### 2.1. Cell Line and Culture Conditions

The human cell line of retinal pigment epithelia ARPE-19 passages 24 to 28 (American Type Culture Collection, Manassas, VA, USA) was grown in DMEM/F12 1:1 medium (Euroclone, Milano, Italy) supplemented with 10% fetal bovine serum and 2 mM glutamine (Euroclone, Milano, Italy) at 37 °C in 5% CO_2_ up to confluence. Afterward, cells were seeded in multi-well plates and cultured for 7–9 days in two different glucose concentrations: 5.6 mM (defined “normal” or NG) and 25 mM (high, HG) before use.

In some experiments, ARPE-19 cells were grown in NG or HG media + 3% FBS in the presence of outer rod segments (OSs, the preparation of which is described in the next section), in amounts corresponding to 10 μg of total protein, for 5.5 h [34]. Subsequently, the growth medium was collected and centrifuged at 20,000× *g* for 5 min to obtain the OS fraction not phagocytosed by the rods, while the ARPE-19 cells were analyzed biochemically to study the crosstalk between the two samples.

### 2.2. Rod OS Isolation

Rod OSs were isolated from retinas extracted from freshly enucleated bovine eyes (obtained from a local slaughterhouse) by a procedure maximizing the OS yield [35]. Briefly, under dim red light, eyecups deprived of vitreous and lens were filled with Mammalian Ringer (0.157 M NaCl, 5 mM KCl, 7 mM Na_2_HPO_4_, 8 mM NaH_2_PO_4_, 0.5 mM MgCl_2_, 2 mM CaCl_2_ pH 6.9 plus protease-inhibitor cocktail (Sigma-Aldrich, St. Louis, MO, USA) and 50 μg/mL ampicillin, for 10 min. Then, floating retinas were cut free of the optic nerve. Afterward, rod OSs were obtained by sucrose/Ficoll continuous gradient centrifugation in the presence of a protease inhibitor cocktail (Sigma–Aldrich, St. Louis, MO, USA) and ampicillin (50 µg/mL), as described in [36]. Before their addition to the culture medium of ARPE-19, the OS samples were split into two aliquots, one of which was kept in the dark, while the other was exposed to ambient light for 30 min. In both cases, rod OSs were supplemented with respiratory substrates (0.6 mM NADH, 20 mM succinate, and 0.1 mM ADP) to trigger their OxPhos. As shown in Appendix A, the OS samples pretreated with the substrates but kept in the dark produced ATP and consumed oxygen without accumulating malondialdehyde (a lipid peroxidation marker). In contrast, rod OSs exposed to ambient light in the presence of the substrates displayed high oxygen consumption but low ATP synthesis, suggesting an OxPhos uncoupling, which favors peroxided lipid accumulation. For these metabolic characteristics, the OSs maintained in the dark were defined as unoxidized (UOx), while the rod OSs exposed to ambient light were called oxidized (Ox).

### 2.3. Oxygen Consumption Rate Evaluation 

Oxygen consumption rates (OCRs) were evaluated by means of an amperometric electrode (Unisense Microrespiration, Unisense A/S, Aarhus, Denmark) in a closed chamber at 37 °C. For the experiment, 2 × 10^5^ ARPE-19 cells were employed after resuspension in phosphate buffer saline (PBS) and permeabilization for 1 min with 0.03 mg/mL digitonin. In total, 10 mM pyruvate plus 5 mM malate or 20 mM succinate were added to stimulate the respiratory pathways composed of Complexes I, III, and IV or Complexes II, III, and IV, respectively [37,38]. In both cases, 0.1 mM ADP was added. For rod OSs, 50 μg of total protein was used, and 0.1 mM NADH and 0.1 mM ADP were employed as respiratory substrates.

### 2.4. Aerobic ATP Synthesis Evaluation

To evaluate the aerobic ATP synthesis by the F_1_F_o_-ATP synthase (ATP Synthase) in ARPE-19 cells, 2 × 10^5^ cells were incubated for 10 min at 37 °C in a medium containing 50 mM Tris-HCl (pH 7.4), 50 mM KCl, 1 mM EGTA, 2 mM MgCl_2_, 0.6 mM ouabain, 0.25 mM di(adenosine)-5-penta-phosphate (an adenylate kinase inhibitor), and 25 μg/mL ampicillin (0.1 mL final volume). The same respiratory substrates employed for OCR evaluation were used [38]. For rod OSs, 50 μg of total protein was used, and 0.1 mM NADH was employed as the respiratory substrate. In both cases, ATP synthesis was induced by 0.1 mM ADP addition. The reactions were monitored by a luminometer (GloMax^®^ 20/20 Luminometer, Promega, Milan, Italia) every 30 s for 2 min, using the luciferin/luciferase chemiluminescent method. ATP standard solutions were used for calibration in a concentration range between 10^−8^ and 10^−5^ M (luciferin/luciferase ATP bioluminescence assay kit CLS II, Roche, Basel, Switzerland) [38]. The ratio between ATP synthesis and OCR was calculated to obtain P/O values to evaluate the OxPhos efficiency. Efficient mitochondria have a P/O value of around 2.5 or 1.5, depending on whether pyruvate/malate or succinate were used as substrates. Conversely, a lower P/O ratio indicates an uncoupled OxPhos in which part of the oxygen is not used for energy production but contributes to the formation of oxidative stress [39].

### 2.5. Evaluation of Lipofuscin Accumulation in ARPE-19 Cells by Confocal Microscopy 

To investigate the pro-oxidative effects of UOx- or Ox-rod OS phagocytosis on ARPE-19 cells, the lipofuscin accumulation was analyzed by confocal microscopy, exploiting its autofluorescence in a 570–620 nm detection channel [40,41]. For this evaluation, the ARPE-19 cells were cultured onto 12 mm glass coverslips in NG or HG media. After the 5.5 h treatments with rod OSs, cells were washed three times with PBS and postfixed with 4% paraformaldehyde (Sigma-Aldrich, St. Louis, MO, USA; Cat# 47608) in PBS for 15 min at 25 °C. Cells were washed three times with PBS and stained with DAPI. Glass coverslips were assembled on microscope glass slides. Fluorescence image (1024 × 1024 × 8 bit) acquisition was performed by a Leica TCS SP2 laser-scanning confocal microscope, using the 488 line of the argon ion laser for excitation through a plan apochromatic oil immersion objective 63× (1.4 NA). The Leica “LAS AF” software package was used for image acquisition and analysis.

### 2.6. Lipoperoxidation Evaluation 

To evaluate oxidative damage, malondialdehyde (MDA) concentration was assessed using the thiobarbituric acid reactive substance (TBARS) assay. This test is based on the reaction of MDA, a breakdown product of lipid peroxides, with thiobarbituric acid (TBA). The TBARS solution contained 26 mM thiobarbituric acid and 15% trichloroacetic acid in 0.25 N HCl. In total, 600 μL of TBARS solution was used, and 50 μg of total protein dissolved in 300 μL of Milli-Q water was added. The mix was then incubated at 95 °C for 60 min. The samples were centrifuged for 2 min at 20,000× *g*, and the supernatants were analyzed spectrophotometrically at 532 nm [42]. 

### 2.7. Antioxidant Enzyme Activity Evaluation

For each assay, 20 μg of total protein was employed. Catalase (CAT) activity was assayed following the decomposition of H_2_O_2_ at 240 nm, using an assay medium composed of 50 mM phosphate buffer (pH 7.0) and 5 mM H_2_O_2_ [38]. Glucose 6-phosphate dehydrogenase (G6PD) activity was assayed spectrophotometrically following NADP reduction at 340 nm, with a solution containing 100 mM Tris-HCl (pH 7.4), 0.5 mM NADP, and 10 mM glucose-6-phosphate [43]. Glutathione reductase (GR) activity was assayed following the oxidation of NADPH with a spectrophotometric analysis at 340 nm. The assay solution contained 100 mM Tris-HCl (pH 7.4), 1 mM EDTA, 5 mM GSSH, and 0.2 mM NADPH [44]. Glutathione peroxidase (GPx) activity was assayed following the decomposition of H_2_O_2_ at 240 nm, using an assay solution containing 100 mM Tris-HCl (pH 7.4), 5 mM H_2_O_2_, and 5 mM GSH. Since H_2_O_2_ is also a substrate of catalases, GPx activity is obtained by subtracting the result of this assay from the catalase activity values [38]. 

### 2.8. Western Blot Analysis 

To assess the expression of LC3, an autophagy marker, GRP78, an unfolding protein response (UPR) marker, and caspase 8, an apoptosis marker, 30 μg of proteins was loaded for each sample to perform denaturing electrophoresis (SDS-PAGE) on 4–20% gradient gels (BioRad, Hercules, CA, USA). The primary antibodies used were anti-LC3 (Novus, Centennial, CO, USA; #NB100-2220), anti-GRP78 (Santa Cruz Biotechnology, Dallas, TX, USA; #sc-1050), anti-caspase 8 (Cell Signaling, Danvers, MA, USA; #9746), and anti-actin (Santa Cruz Biotechnology, Dallas, TX, USA #sc-1616), used as a housekeeping protein. All primary antibodies were diluted 1:1000 in PBS plus 0.15% tween (PBSt). Specific secondary antibodies were employed (Sigma-Aldrich, St. Louis, MO, USA), all diluted 1:10,000 in PBSt. To evaluate the amount of rod OS phagocytosed by ARPE-19 cells, the expression of rhodopsin (the most abundant protein in the rod OS) was quantified in the no-phagocytosed rod OS fraction collected from the growth medium after 5.5 h of incubation with ARPE-19 cells. The primary antibody against Rhodopsin (Sigma-Aldrich, St Louis, MO, USA; #R5403) was diluted 1:5000 in PBSt. A specific secondary antibody was employed (Sigma-Aldrich, St. Louis, MO, USA), diluted 1:10,000 in PBSt. All bands observed were detected and analyzed for optical density using an enhanced chemiluminescence substrate (ECL, BioRad, Hercules, CA, USA), a chemiluminescence system (Alliance 6.7 WL 20M, UVITEC, Cambridge, UK), and UV1D Alliance^TM^ Q9-Series software (UVITEC, Cambridge, UK). 

### 2.9. Statistical Analysis

Results are representative of at least 3 independent experiments. All data were analyzed with GraphPad Prism 8.0 software (GraphPad Software, San Diego, CA, USA). Data were expressed as the mean ± SD and then analyzed using a one-way ANOVA followed by a Tukey’s multiple comparison test. Differences were considered statistically significant if the error probability was *p* < 0.05. 

## 3. Results

Following the hypothesis that the oxidative stress arising in the rod OS after impairment of its respiratory activity due to overwork is involved in the neuronal damage representing the first event in DR pathogenesis, two experimental groups were compared, consisting of ARPE-19 cells grown either in NG or HG media, treated for 5.5 h with rod OSs maintained in the dark (unoxidized, UOx) or exposed to ambient light in the presence of metabolic substrates and ADP (oxidized, Ox) (for more details see Section 2.2 of Materials and Methods). Untreated ARPE-19 cells were used as controls. 

### 3.1. ARPE-19 Cells Increase Aerobic Energy Metabolism Proportionally to Glucose Concentration without Causing an Increase in Oxidative Damage Due to the Activation of Endogenous Antioxidant Defenses

The oxygen consumption rate (OCR), ATP synthesis by F_1_F_o_-ATP synthase, and the relative P/O ratio were evaluated employing pyruvate plus malate or succinate as respiratory substrates to assess changes in the energy metabolism of ARPE-19 cells in NG (5.6 mM) or HG (25 mM) conditions. Increased glucose availability causes a rise in oxygen consumption (Figure 1A) and ATP synthesis (Figure 1B) in the presence of both pyruvate/malate and succinate as respiratory substrates. Interestingly, the evaluation of OxPhos efficiency, in terms of P/O ratio, remains similar under both glucose concentrations (Figure 1C), suggesting that the raising in mitochondrial function does not uncouple respiration and energy production.

OxPhos activity is always associated with ROI production [45,46], leading to increased oxidative damage [47,48]. Nonetheless, ARPE-19 cells grown in HG do not display an increased lipid peroxidation accumulation compared to cells grown in NG glucose, as shown by the similar MDA content (Figure 2A). The absence of additional oxidative damage probably depends on the increased activity of CAT, G6PD, GR, and GPx (Figure 2B–E), enzymes involved in cellular antioxidant defenses.

### 3.2. The Phagocytizing Capacity of ARPE-19 Cells Depends on the Glucose Concentration in the Medium and the Rod Outer-Segment Oxidative State

Rhodopsin quantity was evaluated in the growth medium after ARPE-19 cell incubation with UOx- or Ox-rod OSs (Figure 3) for 5.5 h to verify whether ARPE-19 cells retained phagocytic capacity. Data show that the rhodopsin concentration in the medium of ARPE-19 cells grown in NG after incubation with UOx-rod OSs is significantly lower than in the suspension of rod OSs supplied to the cells, suggesting that the ARPE-19 cells have partly phagocytosed them. This ability further increases when the cells are grown in an HG medium. However, the ARPE-19 phagocytizing capability appears lower when cells are incubated with Ox-rod OSs. This difference could depend on lipid peroxidation accumulation occurring when the OSs are exposed to ambient light in the presence of energy substrates and ADP. This, in turn, causes OxPhos uncoupling, as previously demonstrated [49] and shown in Appendix A.

### 3.3. The Altered ARPE-19 Cell Phagocytosis Capacity Depends on the Accumulation of Oxidative Stress

To understand the effects of UOx- or Ox-rod OS phagocytosis on ARPE-19 cells, the cellular accumulation of lipofuscin, an oxidative stress marker ([44]), has been assessed by confocal microscopy (Figure 4), taking advantage of the autofluorescence features of this byproduct of cellular metabolism. Data show low lipofuscin levels in cells grown without rod OS addition in the presence of both NG and HG media, confirming that when ARPE-19 cells are not challenged with rod OSs, the glucose availability does not cause an increased oxidative damage accumulation. In contrast, when ARPE-19 cells are incubated with UOx-rod OSs, an increment in the lipofuscin concentration has been observed, especially in cells grown in the HG medium, suggesting that the activation of processes related to phagocytosis unmasks the increased risk of oxidative stress linked to the enhanced OxPhos functionality observed in the presence of a high-glucose concentration. However, the most evident lipofuscin accumulation is visible in ARPE-19 cells incubated with Ox-rod OSs, suggesting that the cell oxidative state is influenced by the nature of phagocytosed rod OSs.

Interestingly, despite ARPE-19 cells responding to increased oxidative stress by enhancing the activities of antioxidant enzymes (Figure 5A–D), MDA values increased proportionally to the accumulated lipofuscin (Figure 5E).

### 3.4. The Accumulation of Oxidative Stress due to the Phagocytosis of Ox-rod OSs Causes an Increase in the Expression of Markers of the Unfolding Protein Response and Pro-Apoptotic Signal and Reduces LC3 Expression

To test whether the accumulation of oxidative stress induced by Ox-rod OSs causes changes in the pathways involved in cellular homeostasis, the expression of LC3, an autophagy marker, GRP78, an unfolding protein response (UPR) marker, and the cleaved form of caspase 8, an apoptosis marker, was assessed by Western blot analysis (Figure 6). Data show that LC3 expression increases in ARPE-19 cells treated with UOx-rod OSs compared with untreated cells, with a more pronounced increment when ARPE-19 cells are grown in the HG medium. By contrast, LC3 expression does not change after incubation with Ox-rod OSs under either growth condition compared to the control (Appendix A). Furthermore, despite the different amounts of protein expression, the ratio between the 18 kDa isoform (activated) and the 20 kDa isoform (inactive) remains constant between untreated cells and those incubated with UOx- or Ox-rod OSs when ARPE-19 cells are grown in the NG condition. In contrast, incubation with Ox-rod OSs causes a decrease in the 18/20 kDa ratio in favor of the inactive form in ARPE-19 cells grown in HG.

Assessing the activation of the UPR, the GRP78 expression in ARPE-19 cells grown in NG conditions increases after incubation with rod OSs, reaching the highest level when they were oxidized. The same trend is observed in cells grown in HG, although in the presence of UOx-rod OSs, the GRP78 signal is higher compared to the cells grown in NG conditions.

Finally, cleaved caspase 8, which indicates activation of the protein, follows the same trend of GRP78 in both growth conditions.

## 4. Discussion

This work aimed to investigate a possible reciprocal influence between the pigment epithelium and the apical portion of the rods (rod OS) in establishing a pro-oxidative condition based on excessive oxidative stress production and nutrient availability. For this purpose, ARPE-19 cell cultures and UOx- or Ox-rod OSs were employed. ARPE-19 cells express RPE markers and display physiologically relevant features, such as barrier formation and the ability to phagocytize OSs [33]. In detail, the working hypothesis postulated that phagocytosis of Ox-rod OSs could trigger oxidative damage to the RPE and the outer retina, which is recognized as a principal cause of DR [3,25,50,51]. To mimic DR conditions ex vivo, here ARPE-19 cells were grown under high-glucose (HG, 25 mM) concentrations, which corresponds to an uncompensated diabetic. The cells were challenged once for 5.5 h with either UOx- or Ox-rod OSs and were able to phagocytize them similarly to the in vivo condition. Since rod OS disks express the proteins of the ETC, F_1_F_o_-ATP synthase, and Krebs cycle enzymes [52,53] to meet the bioenergetic needs of phototransduction [28,29,54], rod OSs were exposed to ambient light for 30 min in the presence of respiratory substrates and ADP to promote their endogenous oxidative phosphorylation and to induce rod OS oxidation. Indeed, in these in vitro conditions, rod OSs produce a considerable amount of ROI [29,30], which oxidized the disk membrane as their lipids contain high amounts of long-chain polyunsaturated fatty acids (PUFAs) [19]. The data reported herein show that ARPE-19 cells exhibit energy metabolism supported by oxidative phosphorylation that can adapt based on glucose availability without increasing the accumulation of oxidative damage, which is generally due to the increment of endogenous antioxidant activities (namely, glucose-6-phosphate-dehydrogenase (involved in the generation of NADPH), glutathione reductase (regenerating reduced glutathione), glutathione peroxidase (neutralizing hydrogen peroxide by exploiting reduced glutathione) and catalase (the endoplasmic reticulum enzyme that converts hydrogen peroxide into water and oxygen). In contrast, when ARPE-19 cells phagocytose OS rods, the glucose concentration seems to influence the cellular redox state, as phagocytizing cells grown in HG appear more prone to suffer a permanent oxidative insult than those grown in NG conditions. In other words, the activation of phagocytosis processes seems to unleash a pro-oxidative phenomenon related to excessive glucose. However, the higher oxidative stress accumulation in ARPE-19 cells may depend on the redox state of the phagocytosed objects. In detail, Ox-rod OSs induce more cellular damage than the same unoxidized sample, as shown by increased MDA levels and lipofuscin accumulation. Oxidative damage accumulation can have several consequences for the RPE and, consequently, the retina. Our data show that ARPE-19 cells grown in NG conditions and phagocytosing UOx-rod OSs increase the expression of the proactive (LC3-I, 20 kDa) and active form of LC3 (LC3-II, 18 kDa), probably to eliminate the phagocytosis products, thus causing only a slight increase in markers related to the unfolding protein response (GRP78) and apoptosis activation (caspase 8). In contrast, phagocytosis of Ox-rod OSs enhances LC3 expression and activation, resulting in increased UPR and apoptotic activation, probably triggered by cellular engulfment. The severity of the phenomenon is much more pronounced in ARPE-19 cells grown in the HG medium. In fact, although the expression of the two forms of LC3 was increased in the presence of UOx-rod OSs, ARPE-19 cells grown in high glucose showed increased GRP78 and caspase 8 levels, which are signs of cellular stress. This confirms that the activation of phagocytosis mechanisms in a hyperglycemic environment triggers the activation of cellular pathways that negatively modulate cellular homeostasis. The damage was even more pronounced if ARPE-19 cells were incubated with oxidized rod OSs since LC3 expression did not increase, and the ratio between the active isoform (LC3-II) and the inactive isoform (LC3-I) was unbalanced toward the latter, indicating further cell engulfment, in line with the accumulation of lipofuscin. This alteration leads to a further increase in markers of UPR and apoptosis, suggestive of cellular distress. In other words, a hyperglycemic condition associated with altered energy metabolism could favor an intra and extracellular pro-oxidant microenvironment, triggering a vicious circle involving the RPE and photoreceptors and fueling oxidative stress production. This insight could provide a new perspective to the idea that retinal degeneration depends solely and only on a redox alteration of the RPE. Based on the current data, one could consider the possibility of early oxidative damage of the rod OSs, which affects the RPE in more advanced stages of the disease. This interaction appears consistent with the findings that the early event that causes DR is oxidative damage to outer retinal neurons [24,25]. On the other hand, OS phagocytosis is impaired in several diseases, including diabetes. We recently studied the expression of MerTK, a cell-surface receptor that regulates the phagocytosis of RPE cells, in ARPE-19 cells cultured in HG [14]. The reduced expression of MerTk and increased expression of ADAM9, a protease that causes increased shedding of MerTK, were found. In turn, the decreased expression of MerTK impaired OS binding and internalization and impaired the ability to phagocytose oxidized OSs from the rod [14].

The damage resulting from the lipofuscin granule accumulation in ARPE-19 cells was also demonstrated by the internalization of lipofuscin granules extracted from autopsy human RPE into cultured ARPE-19 cells [55]. The study also showed that lipofuscin granules are present in human RPE in concentrations directly proportional to the age of the subjects and that their presence produces oxidative stress [55]. Notably, relatively few lipofuscin granules accumulate in the RPE of normal subjects, especially in the first decade of life. Therefore, despite the heavy metabolic overload imposed by the large volume of material in the outer segment of the rods to be phagocytosed, degraded, and recycled, there is an effectively regulated mechanism essential for the proper functioning of RPE cells that keeps the undigested material level very low. Nevertheless, some lipofuscin material accumulation occurs with physiological aging and even more so in some pathological conditions, such as diabetic retinopathy.

Similar results were obtained in a study conducted on ARPE-19 cells treated with porcine OSs oxidized by treatment with UV light (254 nm) for three hours [32], electron-dense material accumulation was observed in the swollen lysosomes. However, the porcine OSs used in the cited article were severely oxidized, which renders the model unphysiological. By contrast, our ex vivo model utilizing self-oxidized OSs was able to recapitulate the onset of DR pathology. Despite DR having long been considered a microvascular disease in which vascular damage was thought to precede neuronal damage [56], it is now considered that neuroretinal oxidative damage precedes vascular damage in patients with DR [24]. In addition, the oxidative stress involved in the onset of DR has been localized to the subretinal space [25]. Interestingly, correcting the early subretinal-space oxidative stress could restore vision and mitigate histopathology in diabetic animal models. Imaging studies have shown that oxidative stress can be measured at the interface between the OS and RPE [50], consistent with the existence of a crosstalk between the OS and RPE in the etiology of DR.

## 5. Conclusions

The present data suggest the existence of a crosstalk between the RPE and the OS photoreceptor, which would be decisive in DR pathogenesis. The data also support new findings showing that the primum movens of DR is oxidative damage to outer retinal neurons. The latter would, in turn, cause oxidative damage to the RPE and, ultimately, to the vascular system. The data also show that ARPE-19 cells alone may not be an exhaustive model for studying the role of oxidative stress in the outer retina. We believe that evaluating the interaction between several cell types is necessary. The present experimental model made it possible to highlight a latent phenomenon that could occur in pathological conditions in vivo, opening new horizons for future preventive interventions for DR. Nevertheless, further in vivo experiments in animal models will be necessary to confirm our in vitro data regarding the crosstalk between RPE and rod OSs in oxidative stress generation under hyperglycemic conditions.

## Figures and Tables

**Figure 1 cells-12-02173-f001:**
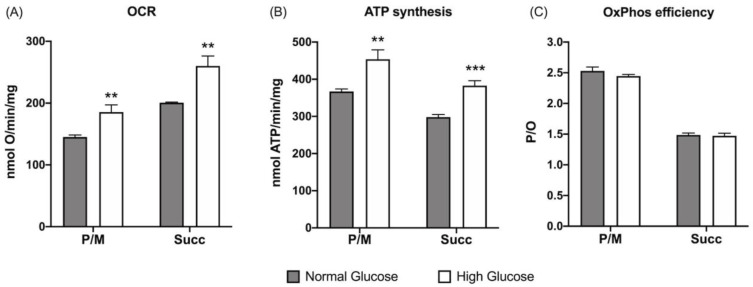
Effect of glucose concentration in growth media on ARPE-19 cell aerobic metabolism: (**A**) oxygen consumption rate (OCR); (**B**) aerobic ATP synthesis; (**C**) P/O values as OxPhos efficiency markers. Aerobic metabolism has been evaluated in the presence of pyruvate plus malate (P/M) or succinate (Succ) to stimulate the pathways of Complexes I, III, and IV and Complexes II, III, and IV, respectively. Gray columns represent data obtained on ARPE-19 cells grown at 5.6 mM glucose (normal glucose, corresponding to the glucose concentration commonly used to grow these cells), and white columns represent the same sample grown in a high-glucose medium (25 mM). Data are expressed as the mean ± SD and are representative of four independent replicates (n = 4). ** and *** indicate an error probability of *p* < 0.01 and 0.001, respectively, between the sample grown in NG or HG media.

**Figure 2 cells-12-02173-f002:**
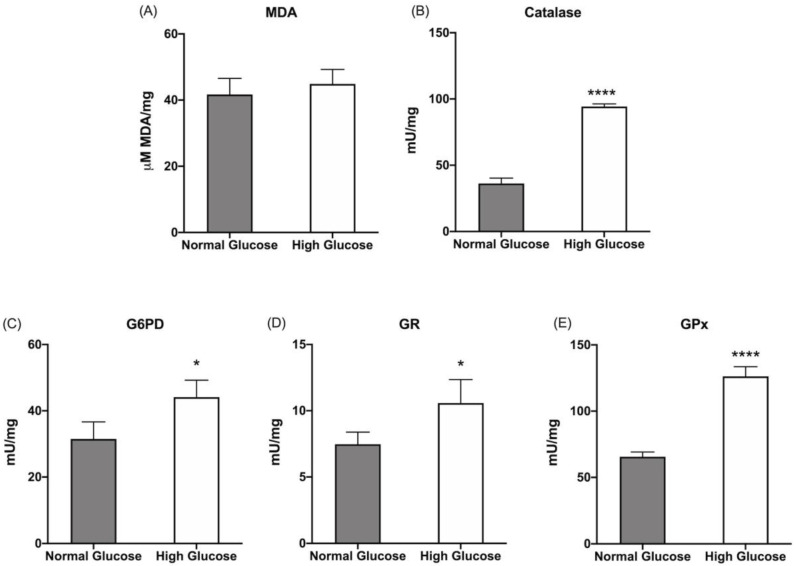
Lipid peroxidation and antioxidant defenses in ARPE-19 cells grown in normal- or high-glucose media: (**A**) malondialdehyde (MDA) concentration as a marker of lipid peroxidation; (**B**) catalase activity; (**C**) glucose 6-phosphate dehydrogenase (G6PD) activity; (**D**) glutathione reductase (GR) activity; (**E**) glutathione peroxidase (GPx) activity. All tested enzymes are involved in cellular antioxidant defenses. Gray columns represent data obtained on ARPE-19 cells grown at 5.6 mM glucose (normal glucose, corresponding to the glucose concentration commonly used to grow these cells), and white columns represent the same sample grown in a high-glucose medium (25 mM). Data are expressed as the mean ± SD and are representative of four independent replicates (n = 4). * and **** indicate a significant difference for *p* < 0.05 or 0.0001, respectively, between the sample grown in NG or HG media.

**Figure 3 cells-12-02173-f003:**
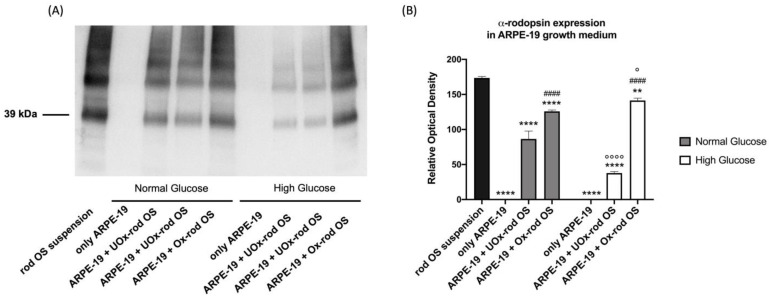
Rhodopsin quantity in ARPE-19 cell growth media as a marker of cell ability to phagocyte unoxidized or oxidized rod OSs. (**A**) Rhodopsin signal in ARPE-19 cell growth media after 5.5 h of incubation with either unoxidized (UOx) or oxidized (Ox) rod OSs. The whole WB signal, including the molecular weight (MW) markers, is reported in Appendix A. (**B**) Densitometric analysis of the rhodopsin monomer signal (39 kDa). The black column represents rhodopsin concentration in the rod OS suspension before incubation; the gray and white columns represent rhodopsin concentration under NG and HG conditions, respectively. Data are expressed as the mean ± SD and are representative of four independent replicates (n = 4). ** and **** indicate an error probability of *p* < 0.01 and 0.0001, respectively, in the rhodopsin concentration before and after the incubation with ARPE-19 cells; #### indicates a *p* < 0.0001 between the ARPE-19 cells incubated with UOx- or Ox-rod OSs; ° and °°°° indicate a *p* < 0.05 and 0.0001, respectively, between the rhodopsin concentration in NG or HG media.

**Figure 4 cells-12-02173-f004:**
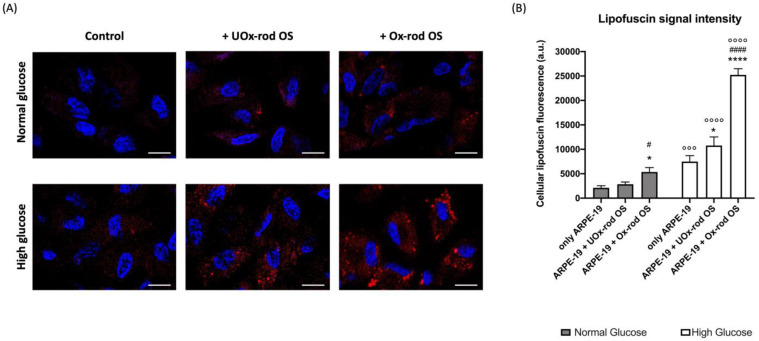
Lipofuscin accumulation in ARPE-19 cells grown in normal- or high-glucose media and incubated with UOx- or Ox-rod OSs. (**A**) Representative confocal images, reporting the cytoplasmic lipofuscin accumulation (red signal) in ARPE-19 cells grown in normal-glucose (NG) and high-glucose (HG) conditions and incubated with either UOx- or Ox-rod OSs. The blue signal corresponds to cells nuclei stained with DAPI. The bar scale corresponds to 10 μm. (**B**) The intensity of the lipofuscin fluorescence signal. Gray columns represent data obtained on ARPE-19 cells grown with NG, and white columns represent the same sample grown in HG conditions. Data are expressed as the mean ± SD and are representative of four independent replicates (n = 4). * and **** indicate a *p* < 0.05 and 0.0001, respectively, between ARPE-19 cells incubated or not with rod OSs; # and #### indicate a *p* < 0.05 and 0.0001 between the ARPE-19 cells incubated with UOx- or Ox-rod OSs; °°° and °°°° indicate a *p* < 0.001 and 0.0001, respectively, between the lipofuscin accumulation in ARPE-19 cells grown in NG or HG media.

**Figure 5 cells-12-02173-f005:**
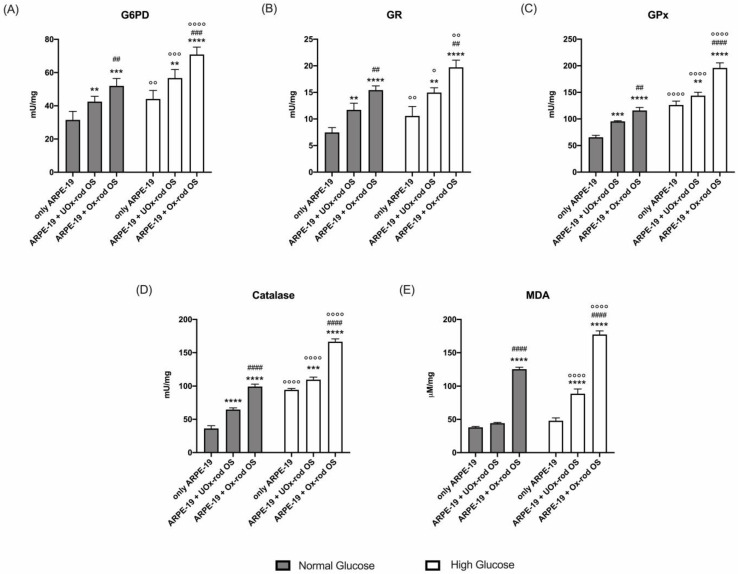
Lipid peroxidation and antioxidant defenses in ARPE-19 cells grown in normal- and high-glucose conditions, after incubation with UOx- or Ox-rod OSs: (**A**) glucose 6-phosphate dehydrogenase (G6PD) activity; (**B**) glutathione reductase (GR) activity; (**C**) glutathione peroxidase (GPx) activity; (**D**) catalase activity; (**E**) malondialdehyde (MDA) concentration as a marker of lipid peroxidation. All tested enzymes are involved in cellular antioxidant defenses. Gray columns represent data obtained on ARPE-19 cells grown in NG, and white columns represent the same sample grown in HG. Data are expressed as the mean ± SD and are representative of four independent replicates (n = 4). **, ***, and **** indicate a *p* < 0.01, 0.001, and 0.0001, respectively, between ARPE-19 cells incubated or not with rod OSs; ##, ###, and #### indicate a *p* < 0.01, 0.001, and 0.0001 between the ARPE-19 cells incubated with UOx- or Ox-rod OSs; °, °°, °°°, and °°°° indicate a *p* < 0.05, 0.01, 0.001, and 0.0001, respectively, between the lipofuscin accumulation in ARPE-19 cells grown in NG or HG media.

**Figure 6 cells-12-02173-f006:**
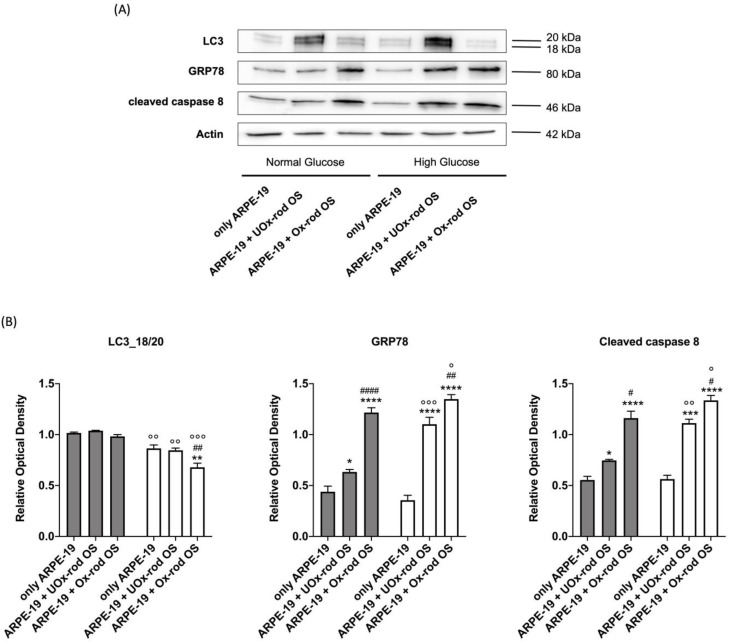
LC3, GRP78, and cleaved caspase 8 expression in ARPE-19 grown in normal- and high-glucose media and incubated with UOx- or Ox-rod OSs. (**A**) WB signals of LC3 (20 and 18 kDa bands), GRP78, cleaved caspase 8, and actin (used as a housekeeping protein) in ARPE-19 cells grown in NG or HG conditions after incubation with either UOx- or Ox-rod OSs. The whole WB signal, including the molecular weight (MW) markers, is reported in Appendix A. (**B**) Densitometric analysis of the ratio between 18 kDa and 20 kDa LC3 bands, GRP78, and cleaved caspase 8 signals. Gray and white columns represent ARPE-19 cells grown in NG or HG media, respectively. Data are expressed as the mean ± SD and are representative of four independent replicates (n = 4). *, **, ***, and **** indicate a significant difference with an error probability of *p* < 0.05, 0.01, 0.001, and 0.0001, respectively, between signals in ARPE-19 cells incubated or not with rod OSs; #, ##, and #### indicate a *p* < 0.05, 0.01, and 0.0001 between the ARPE-19 cells incubated with UOx- or Ox-rod OSs; °, °°, and °°° indicate a *p* < 0.05, 0.01, or 0.001, respectively, between the signal intensity in ARPE-19 cells grown in NG or HG media.

## Data Availability

All the data are contained within the article and the Appendix A.

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
