# Peer review of "Crosstalk between the Rod Outer Segments and Retinal Pigmented Epithelium in the Generation of Oxidative Stress in an In Vitro Model"

_cells, 2023, doi:10.3390/cells12172173_

Round 1
Reviewer 1 Report
The authors use ARPE19 as the model to study the effect of glucose on mitochondrial health in terms of lipid peroxidation and antioxidant protection. The authors connected the phagocytosis with Rhodopsin levels and ultimately to lipofuscin accumulation. The authors could have presented the data more effectively and precisely.
My primary concern is the need for more data on ARPE19 validation in terms of RPE maturity and functionality. If the authors claim to show the relationship between OS and RPE, I would like to see the structural validation of RPE apical processes via SEM and immunostaining for ezrin and phospho ezrin.
The other primary concern I have is the data presentation.
The critical details in the methods sections on a timeline of ARPE19 culture need to be included. The information on the culture format for ARPE19 needs to be included if the cells were seeded on transwells or plastic surfaces.
The number of replicates, n is missing in every experiments.
Minor comments:
Fig3: Western blot lacking any molecular weight ladder. Fails to understand the x-axis for the respective quantification of rhodopsin WB
Fig5, it appears that the relative differences for anti-oxidation markers in high and normal glucose levels are same, it’s just that cells in high glucose levels have higher baseline expression level of the markers and hence in the presence of oxidized and unoxidized OS, its relatively varied.
Fig6. No need to show LC3 18 and LC3 20 blots separately: the ratio of the two works well and molecular ladder is missing.
Author Response
The authors use ARPE19 as the model to study the effect of glucose on mitochondrial health in terms of lipid peroxidation and antioxidant protection. The authors connected the phagocytosis with Rhodopsin levels and ultimately to lipofuscin accumulation. The authors could have presented the data more effectively and precisely.
A1: My primary concern is the need for more data on ARPE19 validation in terms of RPE maturity and functionality. If the authors claim to show the relationship between OS and RPE, I would like to see the structural validation of RPE apical processes via SEM and immunostaining for ezrin and phospho ezrin.
R1: Unfortunately, the short time allowed for this revision is not enough for the requested SEM and immunostaining experiments. Surely, it has been proposed that ezrin, a membrane-organizing actin-plasma membrane crosslinking protein, is involved in the change in membrane tension (un-wrinkling) of the wrinkled phagocyte surface, and ezrin is expressed in the apical ARPE-19 cell membrane. However, human ARPE-19 cell line is widely-used in eye research, as well as general epithelial cell studies of its ability to retain its phagocytic capacity. For example, Becerra and her group utilized bovine rod OS and ARPE_19 cells (doi: 10.1167/iovs.62.2.30) without feeling the need to prove their ability to phagocytize the bovine rod OS they challenged them with. On the other hand, it has been demonstrated that, in ARPE‐19 cells cultured for 5 days, Ezrin localizes preferentially to the apical membrane as in the polarized RPE (doi: 10.1167/iovs.15-19039). Therefore, we may hypothesize that the same occurs in our culture conditions. Also, the results on lipofuscin accumulation indirectly show that cells can phagocytize, in fact the RPE lacks expression of the rhodopsin gene.
Finally, we can show to the Reviewer only the data from Western blotting against rhodopsin in ARPE-19 cells to demonstrate that cells have indeed phagocyted part of rod OS added in the medium (please see the figure below, inserted also as Reviewer Material during manuscript resubmission).
A2: The other primary concern I have is the data presentation.
R2: We apologize for perhaps not allowing the data to be sufficiently clear and we realize that there are many variables at play. Classical abbreviations were used, although they may be a little overwhelming. We tried to further simplify the reading by small changes throughout the text. However, as we did not receive more specific feedback from the Reviewer, it was difficult for us to understand where we needed to change the data presentation.
A3: The critical details in the methods sections on a timeline of ARPE19 culture need to be included. The information on the culture format for ARPE19 needs to be included if the cells were seeded on transwells or plastic surfaces.
R3: As mentioned in the previous version, cells were cultured in multi-well plates for 7-9 days in a medium at two different glucose concentrations (5.6 and 25 mM) before the rod OS incubation. Transwells were not employed.
A4: The number of replicates, n is missing in every experiments.
R4: Indeed, the number of experimental replicates is indicated in each figure legend. However, in the revised version, we have added the words (n=4) to better specify the replicates number.
A5: Minor comments:
Fig3: Western blot lacking any molecular weight ladder. Fails to understand the x-axis for the respective quantification of rhodopsin WB
Fig6. No need to show LC3 18 and LC3 20 blots separately: the ratio of the two works well and molecular ladder is missing.
R5: Regarding the lack of molecular weight markers in Figures 3 and 6, we added two supplementary figures, namely Supplementary Figures 2 and 3, which show the whole WB signal, including the molecular weight (MW) markers. In the case of the actin signal, the MW ladders have been pencil-marked, as the several washes between antibody incubations decolorize them.
We preferred not to add them to the figures in the manuscript as the markers are not visualized in WB using a peroxidase-conjugated secondary antibody capable of recognizing them, in fact, the nitrocellulose membrane has been photographed in ambient light before the chemiluminescent addition. Thus, to assess the molecular weight of the interest band, the image of the membrane and WB are superimposed using the UVITEC software, but this procedure limits the quality and sharpness of the WB signal although not hampering the ability of the instrument to relay the molecular weight of the proteins.
Concerning the densitometry of the signals corresponding to the 18 and 20 KDa bands of LC3, we have removed this figure from Figure 6 in the manuscript, transferring the data to a supplementary figure (Supplementary Figure 4). However, we would like not to remove it completely, because we believe it is important to emphasize that incubation with oxidized rod OS causes a drastic drop in the LC3 expression, indicating a reduced ability to express this autophagy-related protein, other than the percentage of active versus inactive protein.
A6: Fig5, it appears that the relative differences for anti-oxidation markers in high and normal glucose levels are same, it’s just that cells in high glucose levels have higher baseline expression level of the markers and hence in the presence of oxidized and unoxidized OS, its relatively varied.
R6: We completely agree with this reviewer's observation; notably, these data confirm that the high glucose condition makes ARPE less efficient in rod-OS phagocytosis less efficient. In detail, as stated in lines 242-247 of the original version, we think that the increase in antioxidant defenses observed in ARPE-19 grown in high glucose is a response to the increase in aerobic energy metabolism that, however efficient, is always associated with an increase in oxidative stress. Under this assumption, we might expect to observe a higher enhancement in AO defenses in ARPE-19s grown in high glucose and incubated with OS rods (oxidized or not), as simple phagocytosis is itself a stress for ARPEs. However, as pointed out by the reviewer, the increase in AO defenses in ARPEs grown under hyperglycemic conditions is similar to those cultured in normal glucose, suggesting that high glucose levels make ARPEs less responsive to the stress caused by rod OS phagocytosis, causing increased oxidative damage, as shown by both the accumulation of lipofuscin granules and malondialdehyde.

Reviewer 2 Report
Based on large amount of data involving the changes of RPE and photoreceptors on the conditions of NG and HG,a novel theory which suggests that oxidative damage of DR may originate from the photoreceptors and subsequently extend to the RPE。
It is a new perspetive idea to the mechanism of DR and will be elucidated by means of other cells and various conditions.
there is some minor grammar which should be modified, such as line 103~107.
Author Response
Reviewer 2
Based on large amount of data involving the changes of RPE and photoreceptors on the conditions of NG and HG,a novel theory which suggests that oxidative damage of DR may originate from the photoreceptors and subsequently extend to the RPE.
It is a new perspective idea to the mechanism of DR and will be elucidated by means of other cells and various conditions.
We thank the Reviewer for this positive comment.
A1: there is some minor grammar which should be modified, such as line 103~107.
R1: we apologize for the clumsy phrase, which was corrected; the English language was also checked for.
